# Tumour-Agnostic Therapy for Pancreatic Cancer and Biliary Tract Cancer

**DOI:** 10.3390/diagnostics11020252

**Published:** 2021-02-06

**Authors:** Shunsuke Kato

**Affiliations:** Department of Clinical Oncology, Juntendo University Graduate School of Medicine, 2-1-1, Hongo, Bunkyo-ku, Tokyo 113-8421, Japan; katoshun@juntendo.ac.jp; Tel.: +81-3-5802-1543

**Keywords:** pancreatic cancer, biliary tract cancer, targeted therapy, solid tumours, driver mutations, agonist therapy

## Abstract

The prognosis of patients with solid tumours has remarkably improved with the development of molecular-targeted drugs and immune checkpoint inhibitors. However, the improvements in the prognosis of pancreatic cancer and biliary tract cancer is delayed compared to other carcinomas, and the 5-year survival rates of distal-stage disease are approximately 10 and 20%, respectively. However, a comprehensive analysis of tumour cells using The Cancer Genome Atlas (TCGA) project has led to the identification of various driver mutations. Evidently, few mutations exist across organs, and basket trials targeting driver mutations regardless of the primary organ are being actively conducted. Such basket trials not only focus on the gate keeper-type oncogene mutations, such as HER2 and BRAF, but also focus on the caretaker-type tumour suppressor genes, such as BRCA1/2, mismatch repair-related genes, which cause hereditary cancer syndrome. As oncogene panel testing is a vital approach in routine practice, clinicians should devise a strategy for improved understanding of the cancer genome. Here, the gene mutation profiles of pancreatic cancer and biliary tract cancer have been outlined and the current status of tumour-agnostic therapy in these cancers has been reported.

## 1. Introduction

Pancreatic cancer and biliary tract cancer are malignant tumours with the worst prognosis. By 2020, the number of new cases of pancreatic cancer is estimated to be 57,600 and biliary tract cancer would be 42,810, which accounts for 3.2 and 2.4% of all new cancer cases, respectively. In contrast, the estimated deaths due to pancreatic and biliary tract cancers in 2020 would be 47,050 and 30,160, accounting for 7.8 and 5.0% of all cancer-related deaths in the United States [1].

The poor prognoses of pancreatic and biliary tract cancers are attributable to their low sensitivity to systemic chemotherapy. FOLFIRINOX [2] or combination therapy with gemcitabine and nab-paclitaxel [3] is the first-line therapy for unresectable or metastatic pancreatic cancer, and gemcitabine and CDDP (cisplatin) for biliary tract cancer [4]. These treatments are not based on gene mutation profiles, and the median overall survival is less than one year with either treatment.

However, in recent years, with the advancements in the next-generation sequencing (NGS) technology, the genome profiles of a considerable number of tumour specimens have been markedly improved, thereby making it an era wherein therapeutic targets can be explored based on these data, in a so-called tumour-agnostic or histology-agnostic manner [5,6,7,8,9].

Generally, the amount of DNA required for a gene panel test is 10 to 500 ng, and the proportion of tumour cells contained in the sample is important for ensuring the quality of the test [10]. Previously, it was difficult to collect samples for pancreatic cancer and biliary tract cancer, but with the development of endoscopic ultrasonography (EUS) guided biopsy technology, clinically useful samples can now be collected [11,12]. Furthermore, it has been reported that NGS analysis results are obtained from specimens collected by EUS guided biopsy, and that the results are comparable to surgical specimens [13,14]. Therefore, it is expected that gene panel tests will be actively performed in pancreatic cancer and biliary tract cancer.

The cancer panel test introduced in clinical practice has led to the identification of druggable mutations in approximately 10% of the pancreatic cancers and 40% of the biliary tract cancers [15]. Therefore, utilisation of tumour multigene NGS is recommended for the detection of actionable alterations in cholangiocarcinoma [16].

Here, we review the genome profiles of pancreatic cancers and biliary tract cancers and the approaches used for the development of drugs for the indicated actionable mutations.

## 2. Gene Mutation Profiles of Pancreatic Cancers

A considerable number of the gene mutations reported in pancreatic cancer are attributed to *KRAS*, tumour protein p53 (*TP53*), *SMAD4*, and cyclin-dependent kinase inhibitor 2A (*CDKN2A*), and *KRAS* mutations account for approximately 90% of the pancreatic cancers [17,18]. The results of all exon analyses and single-nucleotide polymorphism (SNP) arrays using samples obtained from 24 pancreatic cancer patient indicate that pancreatic cancer exhibits an average of 63 genetic changes, most of which are point mutations and such gene mutations are concentrated in 12 cellular signalling pathways [19].

Furthermore, Biankin, AV et al. [20] performed copy number variation (CNV) analysis using exome sequences and arrays of 142 cases of early pancreatic cancer and reported gene mutations in the molecules of the Axon Guidance signal during the embryonic period in addition to the abovementioned 12 cellular signals (slit guidance ligand 2 (*SLIT2*), roundabout homolog 2 (*ROBO2*) gene mutation, *ROBO1* and *SLIT2* copy number decrease, semaphorin 3A (*SEMA3A*), and plexin-A2 (*PLXNA2*) copy number increase). The decrease in the expression level of these genes caused by the gene structural mutation is found to be associated with a poor prognosis.

Waddell et al. performed whole-genome analysis and CNV analysis of 100 cases of pancreatic cancer and classified pancreatic cancer based on the pattern of chromosomal structural changes into “stable”, “locally rearranged”, “scattered”, and “unstable” types [21]. Among the locally rearranged types, which account for 30% of all pancreatic cancers, there are cases that show amplification of the protooncogenes such as *ERBB2*, *MET*, fibroblast growth factor receptor 1 (*FGFR1*), cell division protein kinase 6 (*CDK6*), phosphoinositide-3-kinase regulatory subunit 3 (*PIK3R3*), and phosphatidylinositol-4,5-bisphosphate 3-kinase catalytic subunit alpha (*PIK3CA*), which are likely to be treated by molecular targeted therapies. Additionally, the unstable type, which accounts for 14% of all pancreatic cancers, contains several abnormal DNA repair-related genes. Furthermore, it was reported that cases with abnormalities in these genes were sensitive to platinum-based drugs.

Raphael et al. performed integrated genomic, transcriptomic, and proteomic profiling of 150 pancreatic ductal adenocarcinomas [6]. They reported that abnormalities in genes of the RAS pathway other than *KRAS* were observed in approximately 60% of the *KRAS* wild-type pancreatic cancers. The proteomic profiling revealed that the TSC/mTOR (mammalian target of rapamycin) signalling pathway showed increased activation in the *KRAS* wild-type pancreatic cancer compared to that in the mutant type, and it was considered as one of the molecular targets of the *KRAS* wild-type pancreatic cancer.

Based on the abovementioned findings, it can be implied that pancreatic cancer is dominated by *KRAS* mutations or RAS pathway genes, but other gene mutations have also been observed that may be targeted for treatment [22]. In the future, it will be necessary to conduct studies to devise strategies for identification of these patients.

## 3. Gene Mutation Profiles in Biliary Tract Cancers

Based on the results of comprehensive genetic analysis of biliary tract cancer, the mutant genes of biliary tract cancer constitute a heterogeneous population consisting of common genetic abnormalities and unique genetic abnormalities in gallbladder cancer, intrahepatic cholangiocarcinoma, and extrahepatic cholangiocarcinoma [7,8,9,23,24,25,26]. 

Li et al. performed a paired analysis with normal specimens of 57 cases of gallbladder cancer and found that gene mutations such as *TP53* (47.1%), *KRAS* (7.8%), and *ERBB3* (11.8%) were frequently observed, and patients with ERBB pathway mutations, which comprised 36.8% of the cohort, had a worse outcome [23]. Roa et al. also reported that patients with gallbladder cancer with HER2 overexpression (*ERBB2* amplification) demonstrated a worse overall survival [24].

Ross et al. analysed 3320 exons of 182 cancer-related genes and 36 introns of 14 genes in 28 cases of intrahepatic cholangiocarcinoma [25]. Structural abnormalities in AT-rich interaction domain 1A (*ARID1A*; 36%), isocitrate dehydrogenase 1/2 (*IDH1/2*; 36%), *TP53* (36%), and induced myeloid leukaemia cell differentiation protein (MCL1; 21%) were observed frequently, and fusion genes of *FGFR2* and neurotrophic tyrosine kinase (*NTRK*) were also reported. 

Nakamura et al. reported results of the whole-exome analysis of 260 cases of biliary tract cancer (145 cases of intrahepatic cholangiocarcinoma, 86 cases of extrahepatic bile duct cancer, and 29 cases of cholangiocarcinoma) [7]. Compared to the intrahepatic cholangiocarcinoma cases, gallbladder cancer and extrahepatic cholangiocarcinoma cases exhibit a higher frequency of gene mutations and an apolipoprotein B mRNA editing enzyme, catalytic polypeptide-like (*APOBEC*)-associated mutational signature [27]. Significant mutations occurred in 32 genes, including *KRAS*, *PIK3CA*, *IDH1*, *NRAS*, guanine nucleotide binding protein, alpha stimulating activity polypeptide (*GNAS*), and *ERBB2*. Among these, *TP53*, *BRCA1*, *BRCA2*, and *PIK3CA* were observed as gene mutations common to the biliary tract cancers. Site-specific gene abnormalities of *EGFR*, *ERBB3*, phosphatase, and tensin homolog (*PTEN*), *ARID2*, mixed-lineage leukaemia protein 2 (*MLL2*), *MLL3*, telomerase reverse transcriptase (*TERT*) promoter mutations were found in gallbladder cancer, *PRKACA* or *PRKACB* fusion, *ELF3*, *ARID1B* in extrahepatic bile duct cancer, and *FGFR2* fusion, *IDH1/2*, *EPHA2*, and BRCA1 associated protein-1 (*BAP1*) in intrahepatic bile duct cancer. *KRAS*, *SMAD4*, *ARID1A*, and *GNAS* mutations were commonly observed in intrahepatic cholangiocarcinoma and extrahepatic cholangiocarcinoma.

Similar to the gene mutation profile of biliary tract cancer, molecular abnormalities related to the RAS pathway are most frequently reported in pancreatic cancer. However, the abnormality of the RAS pathway is only 51.9%, and other pathways or modules include the TGF-β-SWI/SNF-MYC module (40.2%), p53 module (33.9%), epigenetic module (29.3%), and RB-cell cycle module (11.7%). Taken together, it can be inferred that nearly 40% of the indicated genes harbour druggable mutations.

Genetic structural mutations associated with the poor prognosis include deletions of *CDKN2A/B*, *ERBB2*, *TP53*, *KRAS*, *ARID1A*, and deletion of 7q22.1 [8,9,26]. Nakamura et al. also reported that the subgroup that exhibited the gene expression profile similar to that of the immune system and cytokine activation had a poor prognosis [7].

The major druggable gene mutations are shown in Table 1, and the related clinical trials are shown in Table 2.

## 4. Target Therapy for Germline Mutation

### 4.1. BRCA1/2

*BRCA1/2* is a gene that plays a role in DNA repair during homologous recombination [28,29]. In the cells where homologous recombination repair is not realised due to the deficiency in BRCA1/2 functions, the repair mechanism for DNA double-strand breaks causes genomic instability, resulting in carcinogenesis or cell death [30]. It is also known that cells in which BRCA1/2 does not function are extremely sensitive to DNA-damaging therapy [31,32,33,34,35].

The *BRCA1/2* germline mutation has been identified as the causative gene for hereditary breast and ovarian cancer syndrome (HBOC) [36,37,38] but is also one of the causes for development of hereditary pancreatic cancer. *BRCA1* or *BRCA2* germline mutations reportedly increase the relative risk of pancreatic cancer by 2.26-fold (1.26−4.06) and 3.51-fold (1.87−6.58), respectively [36,37]. It has been reported that 25% of the patients with *BRCA1/2* gene germline mutations are at risk of developing pancreatic cancer, and the lifetime risk of developing pancreatic cancer in patients with *BRCA1/2* gene germline mutations is estimated to be 3–8% [39,40,41,42]. Additionally, germline or somatic *BRCA1/2* mutations are associated with 5−9% of the unselected cases of pancreatic cancer [21,43,44]. 

Since abnormalities in the homologous recombination deficiency (HRD)-related genes are observed in pancreatic cancer, clinical trials to verify the efficacy of the poly (ADP-ribose) polymerase (PARP) inhibitors have been conducted in pancreatic cancer patients with germline mutations in *BRCA1/2* [45,46,47,48]. The POLO trial was a double-blind placebo-controlled phase III trial that investigated the efficacy of maintenance therapy with olaparib in patients with *BRCA1* or *BRCA2* germline mutations who did not demonstrate exacerbations with platinum-based first-line therapy. A significant increase in progression-free survival was reported in the olaparib group compared to the placebo group (7.4 vs. 3.8 months) [48].

In contrast to pancreatic cancer, it is unclear whether *BRCA1/2* germline mutation is responsible for the risk of developing biliary tract cancer, but it has been reported that *BRCA1* or *BRCA2* germline mutations increase the relative risk of biliary tract cancer by 1.87-fold (0.59 to 5.88) and 4.97-fold (1.50−16.52), respectively [36,37]. Terashima et al. reported cases of three patients with *BRCA1/2* germline mutations (*BRCA1* mutation: *BRCA2* mutation = 1: 2) among eighty Japanese patients with biliary tract cancer who met ≥1 of the following criteria: (1) hereditary breast and/or ovarian cancer (HBOC) testing criteria modified for biliary tract cancer; (2) revised Bethesda Guidelines (RBGs) modified for biliary tract cancer (modified RBG); (3) familial biliary tract cancer criteria; (4) criteria for young biliary tract cancer [49].

The effectiveness of PARP inhibitors for biliary tract cancer remains unclear, as fewer patients with *BRCA1/2* germline mutations present with biliary tract cancer compared to those presenting with pancreatic cancer. Although effective from a tumour-agnostic perspective, it has been reported that the responsiveness to PARP inhibitors is weak, except for that observed with *BRCA* mutation-associated tumours [50]. Jonsson et al. also reported that HRD scores [51] tended to increase for *BRCA* mutation-associated tumours compared to those for non-BRCA mutation-associated types. The HRD score may be more useful as a biomarker for predicting the efficacy of PARP inhibitors than the germline mutations in the *BRCA1/2* genes. 

### 4.2. DNA Mismatch Repair (MMR) Genes

The functionality of mismatch repair genes (MMR genes) is crucial for the correction of mismatched nucleotides and insertion/deletion loops that are erroneously incorporated into the newly synthesised strand of DNA during replication, using the parental strand of DNA as a template [52]. Functional loss due to structurally mutated or epigenetically inactivated genes results in the accumulation of replication errors [53]. Microsatellite instability (MSI) is a surrogate marker of predisposition to the mutation that results from impaired MMR gene functions [54].

Among the MMR genes, mutL homolog 1(*MLH1*), mutS homolog 2 (*MSH2*), *MSH6*, and *PMS2* are reportedly involved in germline pathogenic mutations in Lynch syndrome [55]. Additionally, since epithelial cell adhesion molecule (*EPCAM*) is located adjacent and upstream to *MSH2* and deletion of *EPCAM* causes decreased expression of MSH2, deletion of the germline *EPCAM* leads to the development of Lynch syndrome [56]. Furthermore, abnormalities in the functions of MMR-related genes other than those of the abovementioned genes may also result in Lynch syndrome [57].

It has been reported that the patients with germline mutations of the MMR gene demonstrate a cumulative risk of 1.31% of developing pancreatic cancer by the age of 50 years and a cumulative risk of 3.68% of developing pancreatic cancer by the age of 70 years, and the risk is estimated to be 8.6 times more that of the general population [58]. Other reports also suggest that MMR gene germline mutations are a risk factor for pancreatic cancer development [59,60]; however, mutations in the MMR genes in hereditary pancreatic cancer are relatively rare [61].

In contrast, in Lynch syndrome, biliary tract cancer is known as one of the high-risk cancers [62,63]. The risk of developing biliary tract cancer in patients with Lynch syndrome with a genetic aetiology is reported to be standardised incidence ratio (SIR) 9.1; 95% CI, 1.1 to 33 [64]. Microsatellite instability (MSI-High) is observed in approximately 5% of the cholangiocarcinoma and extrahepatic cholangiocarcinoma cases, and nearly 10% of the intrahepatic cholangiocarcinoma and ampullary carcinoma cases [65].

The PD-1 antibody pembrolizumab is highly effective against solid tumours characterised by high MSI or deficiency of MMR [66,67]. In the KEYNOTE-158 study [68], which examined the efficacy of pembrolizumab in patients with solid tumours presenting with MSI other than colorectal cancer, the overall response rate was 34.3% and the median PFS was 4.1 months. This study also included patients with pancreatic and biliary tract cancers, with an objective response rate and progression-free survival of 18.2% and 2.1 months in pancreatic cancers, and 40.9% and 4.2 months in biliary tract cancers.

In 2017, the FDA approved the use of pembrolizimab in patients with solid tumours with high MSI/MMR deficiency, and it is reportedly the first case in which diagnosis does not depend on the primary organ as indicators [69]. Tumour mutation burden (TMB), which is defined as the total number of non-synonymous mutations occurring in the exons of tumour cells, also predicts the effect of pembrolizumab. More than 80% of the cases with high MSI are TMB-high, but only 18.3% of the high-TMB cases are also high-MSI [70]. Goodman et al. also reported that tumours with MS-stable/high-TMB phenotypes exhibit better reactivity to PD-1 antibody than tumours with MS-stable/low or intermediate TMB phenotypes. Since tumours with high TMB are more common than the tumours with a high MSI, the effects of the immune checkpoint inhibitors may be expected in more patients.

## 5. Targeted Therapy for Somatic Mutations

### 5.1. ERBB2 (HER2)

*ERBB2,* also known as *HER2*, is a gene of receptor tyrosine kinase that constitutes the *ERBB* family with *EGFR*, *ERBB3*, and *ERBB4* [71]. HER2 lacks a ligand-binding domain and forms heterodimers with the products from other *ERBB* family genes to trigger the activation of the MAPK pathway [72,73].

To date, anti-HER2 monotherapy, such as trastuzumab monotherapy, is less effective against HER2-overexpressed pancreatic cancer [74], but combination therapy with HER2 and other ERBB families is promising.

In a pathway basket trial, nine pancreatic cancer patients showing HER2 overexpression or gene amplification by using fluorescence in situ hybridization (FISH) were treated with the combination therapy of trastuzumab and pertuzumab, which is an HER dimerization inhibitor, and a 22.2% objective response rate was obtained [75]. In the same study, eight patients with biliary tract cancer overexpressing HER2 and three patients with *ERBB2* structural abnormalities (D277Y, S310F, and A775-G776ins YVMA) were treated with pertuzumab plus trastuzumab, and the response rates were 37.5 and 33.3%, respectively [75,76].

Furthermore, DS-8201 is currently expected to be a new anti-HER2 therapy. DS-8201 is an antibody-drug conjugate (ADC) in which an anti-HER2 antibody is bound to the topoisomerase inhibitor deruxtecan. It is effective in breast cancer [77] and gastric cancer [78] showing HER2 overexpression, and the basket trial for solid cancers showing HER2 overexpression, including pancreatic cancer and biliary tract cancer, is currently in progress (NCT04482309). 

### 5.2. IDH1/2

The isocitrate dehydrogenase (IDH) is an enzyme that catalyses the oxidative decarboxylation of isocitrate to α-ketoglutarate (α-KG). Three subtypes were identified, namely, IDH, IDH1, IDH2, and IDH3. Among these subtypes, point mutations in *IDH1* and *IDH2* have been reported in gliomas of grade II and III developed from low-grade glioma [79,80,81,82]. Further, these genes have been reported to be mutated in acute myeloid leukaemia [83] and prostate cancer [84]. The point mutations recurrently observed in tumour tissue are accumulated on *IDH1* R132 and *IDH2* R172. These mutants are gain-of-function mutants, which possess enzyme activities necessary for the conversion of α-ketoglutaric acid (α-KG) into D-2-hydroxyglutarate (2-HG). The 2-HG produced by these mutants contributes to DNA methylation by suppressing the enzymatic activity of histone lysine demethylases and causes oncogene activation and tumour suppressor gene inactivation [85]. *IDH1/2* mutations are biomarkers with a good prognosis in brain tumours but are not related to the prognosis in biliary tract cancer [26].

Ivosidenib, an inhibitor of IDH1 mutants, and enasidenib, an inhibitor of IDH2 mutants, are reportedly effective against relapsed and refractory acute myeloid leukaemia (AML) with *IDH1* and *IDH2* mutations, respectively [86,87,88].

For biliary tract cancer patients with *IDH1* mutations, a double-blind phase 3 controlled trial, ClarIDHy, was conducted to evaluate the efficacy of ivosidenib [89]; it was observed that the progression-free survival of the group treated with ivosidenib was significantly prolonged compared to the placebo group (2.7 vs. 1.4 months). Since crossover was allowed in this study, there was a positive trend in the survival of the ivosidenib group, but the difference was not statistically significant. Although the difference in progression-free survival (PFS) is relatively small, it implies that a new treatment option exists for patients with *IDH1/2* mutations who constitute a certain proportion of biliary tract cancer cases.

### 5.3. BRAF

The *BRAF* gene encodes a serine/threonine kinase, which transmits the MAPK signal [90]. BRAF and CRAF form dimers using the GTP-bound form of RAS activated by receptor tyrosine kinase (RTKs). The dimerised RAFs activate kinase activities of each other and activate downstream MEKs. *BRAF* mutations found in tumours are classified into three types [91]. 

The missense mutation that occurs at codon 600 is classified as Class I and is a mutant that demonstrates a remarkable activity even as a monomer, independent of the RAS activity. In contrast, the kinase activity of the Class II mutation exhibits functionality ranging from intermediate to high levels, and the mutation forms a dimer regardless of the RAS-GTP type and results in MEK activation. The Class III mutation loses kinase activity but binds more tightly than the wild-type BRAF to RAS-GTP, and their binding to and activation of wild-type CRAF is enhanced, leading to an increased ERK signalling (paradoxical activation). Therefore, Class III *BRAF* mutations often co-occur with RTK overexpression or RAS-activating mutations [92,93]. Since paradoxical activation is also caused by BRAF inhibitors [94,95], MEK inhibitors or RTK inhibitors (such as EGFR antibodies) in combination with BRAF inhibitors are useful for solid tumours with such *BRAF* mutations [93,96].

Subbiah et al. reported the effect of the combination therapy of the BRAF inhibitor and the MEK inhibitor in biliary tract cancer cases with the *BRAF* mutation [97]. In the Rare Oncology Agnostic Research (ROAR) basket trial, combination therapy with dabrafenib and trametinib shows good efficacy against biliary tract cancer cases with *BRAF* V600 mutations, an investigator-assessed overall response of 51%, and an independent reviewer-assessed overall response of 47%.

### 5.4. FGFR Fusion Genes

Fibroblast growth factor receptor (FGFR) forms a family of receptors and is categorised into four types with a transmembrane tyrosine kinase structure, namely FGFR1, FGFR2, FGFR3, and FGFR4 [98,99]. Structural abnormalities, such as fusion, amplification, and mutation, are observed in various cancer types, especially *FGFR2* and *FGFR3* [100]. *FGFR2* mutations are found in biliary tract cancer (approximately 7%, especially intrahepatic cholangiocarcinoma) and endometrial cancer (approximately 5%), and *FGFR3* mutations are found in bladder cancer (approximately 25%) [101]. Since *FGFR* structural mutations have been reported recurrently, and in a tumour-agnostic manner, several FGFR inhibitors (antibodies, small molecule compounds) have been developed [102,103,104,105,106,107,108,109,110,111]. These drugs exert a high therapeutic effect on the fusion gene among various structural mutations of *FGFR*. Of these, pemigatinib, an oral inhibitor of FGFR1, FGFR2, and FGFR3, was the earliest FDA-approved drug intended to be used in chemotherapy-resistant cholangiocarcinoma patients with the FGFR2 fusion gene or rearrangement [112].

The FIGHT-202 study was a phase 2 study in which patients with biliary tract cancer who were refractory to prior treatment were grouped according to the presence or absence of *FGFR* abnormality, and the efficacy of pemigatinib in each group was verified [113]. In this study, the objective response rate was 35.5% in the group with *FGFR2* fusion gene or rearrangement, but 0% in the group with other *FGF/FGFR2* mutations and no mutations. The common adverse events were hyperphosphataemia (55%), alopaecia (46%), dysgeusia (38%), diarrhoea (37%), and fatigue (32%). Currently, a phase 3 study for unresectable or metastatic cholangiocarcinoma with *FGFR2* rearrangement is being conducted to compare pemigatinib with gemcitabine plus cisplatin chemotherapy (FIGHT-302; ClinicalTrials.gov, NCT03656536).

At times, resistance mutations are a major clinical problem for molecular-targeted drugs, but a covalent pan-FGFR inhibitor TAS-120 reportedly exerts an antitumour effect against resistance mutations caused by ATP-competitive FGFR inhibitors, such as BGJ398 and Debio 1347 [108].

### 5.5. Rare Fusion Genes

Since the *Bcr-Abl* fusion gene was identified as a driver mutation in chronic myelogenous leukaemia, several fusion genes have been identified in malignancies. In solid tumours, since the *ALK* fusion gene was identified as one of the driver mutations in non-small cell lung cancer [114], it has also been found across cancers of other organs. Additionally, new fusion genes, such as *ROS1*, *RET*, and *NTRK*, have been identified in various solid tumours [115,116]. 

In pancreatic cancer and biliary tract cancer, these fusion genes are rarely reported. As driver mutations tend to exist in a mutually exclusive manner, these fusion genes are found in cases without other driver mutations. It has also been reported that these fusion genes tend to be found in younger patients. There are some case reports showing the effects of inhibitors targeting these genetic abnormalities [117,118,119,120,121]. 

Of these fusion genes, targeted therapy for the *NTRK* fusion gene is the first tissue-agnostic therapy to receive the US FDA approval. [122,123]. The *NTRK* genes constitute a family of nerve growth factor receptors, and *NTRK1*, *NTRK2*, and *NTRK3* are genes encoding the tropomyosin-related kinases TRKA, TRKB, and TRKC, respectively. Larotrectinib [124] and entrectinib [125] have been reported to exert markedly high antitumour effects on solid tumours with *NTRK* fusions (ORR 60−75%) and have already been approved by the FDA and used for treatment.

A noteworthy aspect to be considered in the identification of the fused gene is the presence of a fused gene at the location where driver mutations are not identified in the DNA sequence. As there are restrictions on the probes that can be used in the intron region for the target DNA sequences, targeted hybridization capture-based RNA sequencing or whole transcriptome sequencing may be useful for detecting target genes with unknown fusion partner genes [126].

## 6. Conclusions

As there is an accumulation of a substantial amount of data on gene mutation profiles of solid tumours, tumour-agnostic treatment based on gene mutations continues to progress, and the usage of personalised medicine is becoming widespread in intractable pancreatic cancer and biliary tract cancer. The remaining challenge in this area is the treatment of *KRAS* mutant cancers, because *KRAS* is one of the most frequently mutated genes in pancreatic cancer and biliary tract cancer.

One of the new treatments for the *KRAS* mutant is Sotorasib (AMG510), which shows a specifically irreversible selective inhibitory effect on *KRAS* G12C [127,128]. However, the *KRAS* G12C mutation rate is reported to be high at 13% in lung cancer, and it is only approximately 1% in pancreatic cancer and biliary tract cancer. A novel approach to developing treatments for *KRAS* mutant cancers, such as those targeting synthetic lethality and metabolism, is expected in the future [129].

## Figures and Tables

**Table 1 diagnostics-11-00252-t001:** Druggable mutation profiles of pancreatic cancer and biliary tract cancer.

	Pancreatic Cancer	Biliary Tract Cancer
*BRCA1/2*	~2%	Rare
MMR (MSI-High)	~2%	5~10%
*ERBB2* (amplification)	<5%	1~10%
*BRAF*	<2%	2~5%
*FGFR*	Rare	5~10%
*IDH1/2*	Rare	5~10%
*NTRK, ALK, ROS1, RET*	<1%	<1%

**Table 2 diagnostics-11-00252-t002:** Clinical trials for druggable mutations in pancreatic cancer and biliary tract cancer.

Target	Therapeutic Agent(s)	Selected Clinical Ttials	Status
*gBRCA*1/2 mutation	Olaparib	NCT02184195	Active, not recruiting
MSI-H/dMMR	Pembrolizumab	NCT02628067	Recruiting
*ERBB2* amplification	Trastuzumab and Pertuzumab	NCT02091141	Active, not recruiting
*ERBB2* amplification	T-DXd, DS-8201a	NCT04482309	Recruting
*IDH1/2* mutation	AG-120 (Ivosidenib)	NCT02989857	Active, not recruiting
*BRAF* mutation	Dabrafenib and Trametinib	NCT02034110	Active, not recruiting
*FGFR2* rearrangement	Pemigatinib	NCT03656536	Recruting
*FGFR* mutation/fusion	Erdafitinib	NCT04083976	Recruiting
*FGFR2* fusions/translocations	BGJ398 (Infigratinib)	NCT03773302	Recruiting
*FGFR2* fusion	TAS-120	NCT02052778	Active, not recruiting
*FGFR2* fusion/mutation/amplification	ARQ087 (Derazantinib)	NCT03230318	Recruiting
*FGFR1, 2, 3* alteration	Debio1347	NCT01948297	Terminated
*NTRK* fusion	Larotrectinib	NCT02122913, NCT02637687, NCT02576431	Recruiting
*NTRK1, NTRK2, NTRK3, ROS1,* or *ALK*	RXDX-101 (Entrectinib)	NCT02097810	Completed
*KRAS* G12C	AMG510	NCT03600883	Recruiting

Selected clinical trial identfied on clinicalTrials.gov on 31 January 2021.

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
