# Peer review of "Tumour-Agnostic Therapy for Pancreatic Cancer and Biliary Tract Cancer"

_diagnostics, 2021, doi:10.3390/diagnostics11020252_

Round 1

Reviewer 1 Report

This is a very interesting and comprehensive review concerning gene mutations and agnostic therapy for pancreatic and biliary cancer. The manuscript is exhaustive and well written.

I have only one point for the Authors to be added to the introduction of the study.

Recent advances in endoscopic ultrasound-guided tissue acquisition have led to the possibility to perform NGS also on biopsy samples. This point is very important considering that biopsy specimen is the only tissue available for molecular diagnostics in patients with locally advanced or metastatic pancreatic cancer that will never undergo surgical resection. Please add this point in the discussion section. I suggest referring to:

- Di Leo M, et al. Dig Liver Dis. 2019 Sep;51(9):1275-1280. A prospective study reporting the histological yield of the new fork-tip needle.

- Crinò SF, et al. Gastrointest Endosc. 2020 Sep;92(3):648-658.e2 This is a prospective study reporting the histological yield of new fine-needle biopsy needles.

- Dreyer SB, et al. Chin Clin Oncol. 2019 Apr;8(2):16. A large study reporting the feasibility of NGS sequencing on endoscopic ultrasound-acquired specimens.

- Larghi A, et al. Pancreatology. 2020 Jun;20(4):778-780. This is the only available study that confirms the reliability of mutations found on biopsy specimens compared with surgical specimens.

Reviewer 2 Report

Shunsuke Kato in his review article titled “Tumour-Agnostic agnostic therapy for pancreatic cancer and biliary tract cancer” systematically reviewed the gene mutation profiles of pancreatic cancer and biliary tract cancer outlining the current status of tumour -agnostic therapy in these cancers. Heh as first given the mutation profiles in both the cancers and then reviewed the strategies for therapy of germline and somatic mutations separately.

It would be nice if the author provide separate tables indicating the somatic mutations, germline mutations in these cancers and also a separate table with all the trials that have been finished with an outcome and ongoing trials.
